# OpenReview forum: "Automatic Model Selection with Large Language Models for Reasoning"
_EMNLP/2023/Conference — EMNLP 2023 Findings_

### Official Review · Reviewer_eeSr · 2023-08-08

**Soundness:** 4

**Excitement:**

4: Strong: This paper deepens the understanding of some phenomenon or lowers the barriers to an existing research direction.

**Paper Topic And Main Contributions:**

This paper proposes a method for effectively combining two different inference methods: Chain-of-Thought (CoT) with an intermediate inference step in natural language and Program-Aided-Language Models (PAL) with a processing step in programmed language. Main idea is letting the LLM itself dynamically choose one of the two methods.

Proposed method improved performance in most cases with Codex, ChatGPT, and GPT-4 LLMs on 8 inference datasets. Further performance gains were achieved by combining with other prompting methods, and new state-of-the-art performance was achieved on GSM8K, an arithmetic reasoning task.
Furthermore, analysis sections provide several interesting insights.

**Questions For The Authors:**

- A. Are a few-shot examples given for each of the steps of CoT execution, PAL execution and model selection?
- B. How many examples are prepared for each task and how many of them are used during inference?
- C. Is there a special procedure for creating examples (e.g., manually selecting samples from training data not randomlly)?
- D. Do you think that selecting a model from three or four options, including 'CoT' and CCoT, would improve performance?

**Reasons To Accept:**

- Proposed a simple, intuitive, and effective prompting method.
- The proposed method is complementary to the self-consistency (a prompting technique) and can be combined with it to further improve the performance.
- Achieved new state-of-the-art performance on GSM8K, an arithmetic reasoning task
- Provided interesting insights:
    - Even though CoT and PAL have similar performance, the performance difference in each sample is actually high, and there is room for performance improvement through combining them with the model selection.
    - The combination of CoT and PAL is a more successful in the model selection than other combinations.
    - The three LLMs used tended to prefer the first option in the model selection

**Reasons To Reject:**

- Seems to be a lack of explanation about the experimental setup (Question A-C)

**Reproducibility:**

3: Could reproduce the results with some difficulty. The settings of parameters are underspecified or subjectively determined; the training/evaluation data are not widely available.

**Reviewer Confidence:**

3: Pretty sure, but there's a chance I missed something. Although I have a good feel for this area in general, I did not carefully check the paper's details, e.g., the math, experimental design, or novelty.

**Typos Grammar Style And Presentation Improvements:**

- In Section 2.1, it is better to mention the weaknesses of the CoT too, so that the CoT and PAL can be contrasted more neatly.
- l.225: period is not needed after the footnote number 1.

---

> ### Author Rebuttal · Authors · 2023-08-29
>
> We appreciate your constructive feedback.
> > A. Are a few-shot examples given for each of the steps of CoT execution, PAL execution and model selection?
>
> We generate the whole reasoning chain of CoT and PAL respectively first, and then use our method to select which method to adopt. The few-shot examples of the selection module contains examples of the whole reasoning chains.
>
> > B. How many examples are prepared for each task and how many of them are used during inference?
>
> We follow prior works. For arithmetic reasoning tasks, we employ an 8-shot approach for Codex and a 5-shot example for ChatGPT and GPT-4. For date understanding tasks, a 6-shot example is utilized for all three backbones. This standardization also applies in the inference phase.
>
> > C. Is there a special procedure for creating examples (e.g., manually selecting samples from training data not randomlly)?
>
> (Below) Accuracy on GSM8K with the same backbone model. We use different few-shot examples here. The last two rows are the results in our paper.
> | Base 1           | Base 2           | Accuracy  |
> | ---------------- | ---------------- | --------- |
> | [ChatGPT] CoT 80.8 | [ChatGPT] PAL 79.2 | 82.4±0.22 |
> | [GPT-4] CoT 94.6   | [GPT-4] PAL 94     | 95.6±0.05 |
> | [ChatGPT] CoT 80.8 | [ChatGPT] PAL 79.2 | 82.6 |
> | [GPT-4] CoT 94.6   | [GPT-4] PAL 94     | 95.6 |
>
> We don’t have a special procedure for creating examples to emphasize the simplicity and effectiveness of our method. We follow the examples used in the baseline models. Additionally, to validate the robustness of our method, we conducted supplementary experiments using 5 randomly sampled questions from the GSM8K training set as few-shot examples. The results indicate that our method maintains its generalizability and is not overly sensitive to prompt variations.
>
>
> > D. Do you think that selecting a model from three or four options, including 'CoT' and CCoT, would improve performance?
>
> (Below) Accuracy with more options on GSM8K with ChatGPT.
> | Base1    | Base2    | Base3     | Accuracy | Success Selection | Upper Bound |
> | -------- | -------- | --------- | -------- | ----------------- | ----------- |
> | CoT 80.8 | PAL 79.2 | CoT' 79.2 | 81.4     | 58.2              | **92.4**    |
> | CoT 80.8 | PAL 79.2 | -         | **82.6** | **60.4**              | 89.4        |
>
> We experimented with expanding the selection options to include CoT, PAL, and CoT'. As shown above, it also achieved performance gains, but the improvement was not that significant as two choices settings. We speculate that the increased number of choices might complicate the selection process, thereby lowering the success selection rate.
> But it's worth noting that a higher upper bound of these three models suggests the potential for improvement. Extending the selector to accommodate more models could indeed be a promising avenue for future research to optimize performance.
>
> > In Section 2.1, it is better to mention the weaknesses of the CoT too, so that the CoT and PAL can be contrasted more neatly.
>
> CoT is weak in handling mathematical calculations and intricate logical relationships.In the revised version of the paper, we will incorporate a more detailed comparison between CoT and PAL.
>
> > l.225: period is not needed after the footnote number 1.
>
> Thank you for pointing out this formatting oversight. The period after footnote number 1 will be removed in the revised manuscript.

---

### Official Review · Reviewer_wWYq · 2023-08-09

**Soundness:** 2

**Excitement:**

3: Ambivalent: It has merits (e.g., it reports state-of-the-art results, the idea is nice), but there are key weaknesses (e.g., it describes incremental work), and it can significantly benefit from another round of revision. However, I won't object to accepting it if my co-reviewers champion it.

**Paper Topic And Main Contributions:**

The paper amalgamates two reasoning approaches, CoT and PAL, and employs a third Large Language Model (LLM) for selecting between the two methods' outcomes. Ultimately, the authors test their approach on various reasoning datasets, achieving several state-of-the-art (SOTA) results. Additionally, they provide an extensive theoretical analysis to elucidate the performance enhancement.

**Questions For The Authors:**

A. I am particularly interested in understanding the overhead introduced by the method. Would its implementation necessitate a substantial increase in processing time?

B. For PAL method's reliance on LLM-generated programs, what is the accuracy of these programs? As per my knowledge, even advanced LLMs struggle with achieving high Pass@1 scores on benchmarks like MBPP.

**Reasons To Accept:**

+ The notion of utilizing large models for selection is innovative and effectively leverages the comprehension capabilities of these models.
+ Thorough theoretical analysis substantiates the method's advantages.
+ The paper is well-written and easily comprehensible.

**Reasons To Reject:**

- The improvements observed on GPT-4 are marginal, with most gains being minimal; it remains to be seen whether the slight benefits from the multiple reasoning in this approach justify the additional complexity compared to traditional single-pass methods(CoT/PAL).
- The authors have not conducted experiments or comprehensive analyses on the extra time and cost overhead introduced by their method.
- While experiments with proficient closed-source models like ChatGPT and CodeX are conducted, it is suggested to supplement the analysis with experiments on open-source models like Llama to assess the method's generalizability.

**Reproducibility:**

3: Could reproduce the results with some difficulty. The settings of parameters are underspecified or subjectively determined; the training/evaluation data are not widely available.

**Reviewer Confidence:**

4: Quite sure. I tried to check the important points carefully. It's unlikely, though conceivable, that I missed something that should affect my ratings.

---

> ### Author Rebuttal · Authors · 2023-08-29
>
> Thanks for the insightful feedback!
> > The improvements observed on GPT-4 are marginal, with most gains being minimal; it remains to be seen whether the slight benefits from the multiple reasoning in this approach justify the additional complexity compared to traditional single-pass methods(CoT/PAL).
>
> We reduce the error rate from 4.2 to 3.2 (95.8→96.8) on GSM8K, the error reduction is high at 23.8% and establish a new state-of-the-art. It is hard to improve on methods that already achieve a very high score while it is relatively easier to obtain bigger improvements with weaker baselines. Some concurrent work achieved similar performance gains at this scale, but used more computation cost. For example, the [PHP approach](https://arxiv.org/pdf/2304.09797.pdf) requires 40 samples to achieve an improvement from 95.5% to 96.5% on GSM8K. In contrast, our approach not only attains a new state-of-the-art but does so with less computational cost.
>
> > The authors have not conducted experiments or comprehensive analyses on the extra time and cost overhead introduced by their method.
>
> (Below) Computation cost with **ChatGPT** on GSM8K. #Tokens = #Input Tokens + #Generated Tokens. Cost is calculated based on OpenAI Pricing.
>
> | Method      | Accuracy | Cost(USD) | #Tokens    | #Gen. Tokens |
> | ----------- | -------- | --------- | ---------- | ------------ |
> | CoT+SC@5    | 85.4     | 11.23     | 7.19M      | 906K         |
> | CoT+SC@10   | 86.7     | 22.46     | 14.37M     | 1.81M        |
> | CoT+SC@15   | 87.4     | 33.69     | 21.56M     | 2.72M        |
> | CoT+SC@20   | 87.4     | 44.92     | 28.74M     | 3.62M        |
> | PAL+SC@5    | 80.9     | 13.88     | 9.04M      | 632K         |
> | PAL+SC@10   | 82.6     | 27.76     | 18.09M     | 1.27M        |
> | PAL+SC@15   | 82.4     | 41.64     | 27.13M     | 1.90M        |
> | PAL+SC@20   | 82.7     | 55.52     | 36.18M     | 2.54M        |
> | **Ours @5** | **88.2** | **29.29** | **18.99M** | **1.61M**    |
>
>
> (Below) Computation cost with **GPT-4** on GSM8K. #Tokens = #Input Tokens + #Generated Tokens. Cost is calculated based on OpenAI Pricing.
>
> | Method      | Accuracy | Cost(USD) | #Tokens    | #Gen. Tokens |
> | ----------- | -------- | --------- | ---------- | ------------ |
> | CoT+SC@5    | 95.6     | 183       | 5.21M      | 891K         |
> | CoT+SC@10   | 95.5     | 367       | 10.43M     | 1.79M        |
> | CoT+SC@15   | 95.8     | 550       | 15.64M     | 2.68M        |
> | CoT+SC@20   | 95.8     | 733       | 20.85M     | 3.57M        |
> | PAL+SC@5    | 94.7     | 173       | 5.13M      | 633K         |
> | PAL+SC@10   | 95.4     | 346       | 10.27M     | 1.27M        |
> | PAL+SC@15   | 95.5     | 519       | 15.4M      | 1.90M        |
> | PAL+SC@20   | 95.8     | 692       | 20.53M     | 2.53M        |
> | **Ours @5** | **96.5** | **386**   | **11.33M** | **1.54M**        |
>
>
> Our approach offers a more efficient computational cost, especially when multi-path inference is considered. Our computational costs are marginally higher than those for CoT+SC@10 and PAL+SC@10 but are more efficient than CoT@15 and PAL@15. Moreover, our method surpasses the performance of CoT@20 and PAL@20, highlighting its cost-efficiency.
>
> Regarding inference time, our method allows parallel execution of CoT and PAL, adding only a single step to the overall inference time. For multi-pass inference, we could use batch API requests to concurrently receive results for multiple CoT and PAL pathways. This parallelization reduces the time required for inference.
>
> > While experiments with proficient closed-source models like ChatGPT and CodeX are conducted, it is suggested to supplement the analysis with experiments on open-source models like Llama to assess the method's generalizability.
>
> We fully recognize the importance of this aspect. Due to time constraints and limited computational resources, we are not able to perform experiments on Llama during the rebuttal period. However, we are committed to extending our evaluation to open-source models and will update the paper with these results as soon as possible.
>
> > For PAL method's reliance on LLM-generated programs, what is the accuracy of these programs? As per my knowledge, even advanced LLMs struggle with achieving high Pass@1 scores on benchmarks like MBPP.
>
> The benchmarks on GSM8K are not directly comparable to those on MBPP. GSM8K mainly focuses on simple calculations that often do not require intricate programming, thereby not fully reflecting the LLM's program generation capabilities. Furthermore, as indicated in Gao et al. (2023), LLMs have demonstrated the ability to generate executable programs as part of their intermediate reasoning steps.
>
> | Backbone | Accuracy |
> | -------- | -------- |
> | Codex    | 71.5     |
> | ChatGPT  | 79.2     |
> | GPT-4    | 94.0     |
> The accuracy of generated programs on GSM8K is shown above. For full results, please refer to table 1 in our paper.

---

### Official Review · Reviewer_C5gG · 2023-08-10

**Soundness:** 2

**Excitement:**

3: Ambivalent: It has merits (e.g., it reports state-of-the-art results, the idea is nice), but there are key weaknesses (e.g., it describes incremental work), and it can significantly benefit from another round of revision. However, I won't object to accepting it if my co-reviewers champion it.

**Paper Topic And Main Contributions:**

The paper delves into the integration of two established models, CoT and PAL, leveraging an LLM for model selection. The main contributions include a theoretical analysis supporting the feasibility of this model combination and empirical results demonstrating performance improvements across various datasets with different backbone LLMs.

**Questions For The Authors:**

A. I would appreciate further insights into the computational overhead associated with the proposed method. Does its implement result in a significant augmentation of processing duration?

B. Could you provide more details on the hyperparameter tuning process?  How were the optimal values determined, especially for temperatures in the model selection process?

C. Given the sensitivity of the selector to prompts, how do you envision enhancing its robustness for broader applications?

**Reasons To Accept:**

+ The paper provides atheoretical analysis to support the feasibility of the proposed model combination.
+ The empirical results are robust, showcasing performance improvements across multiple datasets and with various LLM backbones.
+ The paper is well written and easy to follow.

**Reasons To Reject:**

- The paper primarily utilizes existing models such as CoT and PAL without introducing notable innovations. Its contribution appears to be an application of known methods rather than presenting a transformative approach or fresh insights.
- The paper heavily leans on existing models without introducing significant novel techniques or modifications. This might position the paper as offering incremental improvements rather than groundbreaking contributions.
- The paper's reliance on in-context learning and self-evaluation abilities of LLMs might make the selector sensitive to prompts, potentially limiting its broader applicability.
- While the paper acknowledges LLM biases, it doesn't seem to propose any concrete solutions or mitigation strategies

**Reproducibility:**

3: Could reproduce the results with some difficulty. The settings of parameters are underspecified or subjectively determined; the training/evaluation data are not widely available.

**Reviewer Confidence:**

4: Quite sure. I tried to check the important points carefully. It's unlikely, though conceivable, that I missed something that should affect my ratings.

---

> ### Author Rebuttal · Authors · 2023-08-29
>
> Thank you for the insightful feedback.
> > The paper primarily utilizes existing models such as CoT and PAL without introducing notable innovations. Its contribution appears to be an application of known methods rather than presenting a transformative approach or fresh insights.
>
> To the best of our knowledge, our work is the first work that shows LLM can be used to combine different systems with different strengths. The method is highly effective and achieves state-of-the-art on GSM8K, while being simple to implement. The simplicity and effectiveness of our method does not mean that our work is not novel. Simplicity should be treated as a feature, not a bug, just like other effective deep learning methods. That being said, novelty is a subjective matter and different researchers have different tastes.
>
> > The paper's reliance on in-context learning and self-evaluation abilities of LLMs might make the selector sensitive to prompts, potentially limiting its broader applicability.
>
> (Below) Accuracy on GSM8K with the same backbone model. We use different few-shot examples here. The last two rows are the results in our paper.
> | Base 1           | Base 2           | Accuracy  |
> | ---------------- | ---------------- | --------- |
> | [ChatGPT] CoT 80.8 | [ChatGPT] PAL 79.2 | 82.4±0.22 |
> | [GPT-4] CoT 94.6   | [GPT-4] PAL 94     | 95.6±0.05 |
> | [ChatGPT] CoT 80.8 | [ChatGPT] PAL 79.2 | 82.6 |
> | [GPT-4] CoT 94.6   | [GPT-4] PAL 94     | 95.6 |
>
> (Below) Accuracy on GSM8K with different backbones w/ and w/o explanations.
> | Backbone | Explanation | Accuracy |
> | -------- | ----------- | -------- |
> | Codex    | w/o exp     | **74.7** |
> | Codex    | w/ exp      | 74.6     |
> | ChatGPT  | w/o exp     | 81.8     |
> | ChatGPT  | w/exp       | **82.6** |
> | GPT-4    | w/o exp     | 95.5     |
> | GPT-4    | w/ exp      | **95.6** |
>
> We conducted additional experiments, sampling 5 questions from the GSM8K training dataset as few-shot examples. We performed these experiments 5 and 3 times for ChatGPT and GPT-4, respectively. The results show that our method is robust to prompt variations.
>
> > While the paper acknowledges LLM biases, it doesn't seem to propose any concrete solutions or mitigation strategies
>
> The order bias has also been observed in concurrent work [Zheng2023Judging](https://arxiv.org/pdf/2306.05685.pdf). To address this, we tried a probability calibration method as described in [Zhao2021Calibrate](https://arxiv.org/pdf/2102.09690.pdf). The accuracy of PAL-CoT selection on GSM8K slightly decreased from 74.7% to 74.5% with Codex. We acknowledge that eliminating LLM biases is an important concern within the research community, which could be an interesting future direction.
>
> > Does its implement result in a significant augmentation of processing duration?
>
> (Below) Computation cost with **ChatGPT** on GSM8K. #Tokens = #Input Tokens + #Generated Tokens. Cost is calculated based on OpenAI Pricing.
>
> | Method      | Accuracy | Cost(USD) | #Tokens    | #Gen. Tokens |
> | ----------- | -------- | --------- | ---------- | ------------ |
> | CoT+SC@5    | 85.4     | 11.23     | 7.19M      | 906K         |
> | CoT+SC@10   | 86.7     | 22.46     | 14.37M     | 1.81M        |
> | CoT+SC@15   | 87.4     | 33.69     | 21.56M     | 2.72M        |
> | CoT+SC@20   | 87.4     | 44.92     | 28.74M     | 3.62M        |
> | PAL+SC@5    | 80.9     | 13.88     | 9.04M      | 632K         |
> | PAL+SC@10   | 82.6     | 27.76     | 18.09M     | 1.27M        |
> | PAL+SC@15   | 82.4     | 41.64     | 27.13M     | 1.90M        |
> | PAL+SC@20   | 82.7     | 55.52     | 36.18M     | 2.54M        |
> | **Ours @5** | **88.2** | **29.29** | **18.99M** | **1.61M**    |
>
>
> (Below) Computation cost with **GPT-4** on GSM8K. #Tokens = #Input Tokens + #Generated Tokens. Cost is calculated based on OpenAI Pricing.
>
> | Method      | Accuracy | Cost(USD) | #Tokens    | #Gen. Tokens |
> | ----------- | -------- | --------- | ---------- | ------------ |
> | CoT+SC@5    | 95.6     | 183       | 5.21M      | 891K         |
> | CoT+SC@10   | 95.5     | 367       | 10.43M     | 1.79M        |
> | CoT+SC@15   | 95.8     | 550       | 15.64M     | 2.68M        |
> | CoT+SC@20   | 95.8     | 733       | 20.85M     | 3.57M        |
> | PAL+SC@5    | 94.7     | 173       | 5.13M      | 633K         |
> | PAL+SC@10   | 95.4     | 346       | 10.27M     | 1.27M        |
> | PAL+SC@15   | 95.5     | 519       | 15.4M      | 1.90M        |
> | PAL+SC@20   | 95.8     | 692       | 20.53M     | 2.53M        |
> | **Ours @5** | **96.5** | **386**   | **11.33M** | **1.54M**        |
>
> For the computation cost, we actually reduce computational overhead, when looking at multi-path inference. As illustrated in our results, our approach's computational cost is slightly higher than that of CoT+SC@10 and PAL+SC@10, but lower than CoT@15 and PAL@15. But our approach even outperforms CoT@20 and PAL@20, underscoring its efficiency.
>
> For the inference time, we can run the inference of CoT and PAL in parallel, which only increases the inference time by one step. For multiple passes inference, we can send batch requests for each API call, which can get multiple CoT and PAL results at once and make the selection in parallel, saving the inference time greatly. Notably, our method with 5 paths performs better than CoT+SC@20, with a lower computation cost and shorter inference time.
>
> > Could you provide more details on the hyperparameter tuning process? How were the optimal values determined, especially for temperatures in the model selection process?
>
> In our study, we found that our method is robust to hyperparameters and did not tune them extensively. For the selection LLM, we used a temperature of 0 for a deterministic result. Although further tuning hyperparameters would potentially lead to improved results, we avoided further tuning and emphasized the simplicity and effectiveness of our method. For CoT and PAL, we followed prior works for hyperparameters and prompts.

---

### Official Review · Reviewer_iYd8 · 2023-08-11

**Soundness:** 3

**Excitement:**

4: Strong: This paper deepens the understanding of some phenomenon or lowers the barriers to an existing research direction.

**Paper Topic And Main Contributions:**

The paper delves into the challenge of leveraging the capabilities of various reasoning methods, with a focus on different prompting methods.

The primary contributions are as follows:

- The paper offers insights into how different prompting methods are adept at handling diverse reasoning problems. Specifically, the natural language-based CoT and the programming language-based PAL exhibit distinct distributions of correctly solved problems.

- The authors advocate for the use of LLMs for model selection. Through a thorough theoretical analysis, they identify two primary factors that significantly influence their proposed method.

- The introduced approach achieves state-of-the-art accuracies of 96.8% on GSM8K and 93.7% on SVAMP.

**Questions For The Authors:**

Question A: Regarding the PAL results, are the final outcomes integrated into the selection model, or is it limited to the generated program alone? Does this model exclusively employ its "self-debug" feature for evaluating these results?

Question B: The authors selected CoT and PAL as their primary models, grounding their choice on the assertion that these models "represent unique reasoning mechanisms via natural language and programming language" respectively. Is there an exhaustive statistical evaluation detailing the performance of both methodologies? Could you delineate instances or scenarios where CoT demonstrates superiority? Conversely, under which conditions does PAL showcase optimal performance?

**Reasons To Accept:**

- The paper presents a comprehensive theoretical analysis that quantifies the error rates of base methods and delineates the conditions under which the proposed methodology enhances performance effectively.

- The paper is articulately written, ensuring clarity and ease of understanding. Furthermore, the empirical evidence robustly supports the theoretical claims, achieving state-of-the-art results on GSM8K and SVAMP.

**Reasons To Reject:**

- While the paper emphasizes "model selection", the experimental design primarily addresses "prompting method selection", with both base methods and the selection model leveraging the same LLM backbone. Consequently, the proposed approach can be perceived as a novel prompting strategy, sequentially generating CoT and PAL results using the backbone LLM. It would be beneficial to conduct further experiments with genuinely distinct models to truly harness their respective strengths, such as integrating CodeX and ChatGPT with GPT-4.

- The proposed method demonstrates effectiveness, but it necessitates thrice the computational and inferential resources for a single problem. The "model selection" issue as addressed in this paper seems to be a variant of the self-consistency methods, a topic that has been extensively explored. Prioritizing the selection of prompt strategies based on a specific problem might present a more significant research avenue.

- The motivation behind the paper stems from the notion that CoT and PAL excel in different task types. However, a thorough analysis and explicit examples highlighting the disparities between these two methods are conspicuously absent. Furthermore, there's a need for elucidation on why these methods showcase different correct distributions within the experimental section.

**Reproducibility:**

4: Could mostly reproduce the results, but there may be some variation because of sample variance or minor variations in their interpretation of the protocol or method.

**Reviewer Confidence:**

3: Pretty sure, but there's a chance I missed something. Although I have a good feel for this area in general, I did not carefully check the paper's details, e.g., the math, experimental design, or novelty.

---

> ### Author Rebuttal · Authors · 2023-08-29
>
> We thank the reviewer for their insightful comments.
>
> > While the paper emphasizes "model selection", the experimental design primarily addresses "prompting method selection", with both base methods and the selection model leveraging the same LLM backbone. Consequently, the proposed approach can be perceived as a novel prompting strategy, sequentially generating CoT and PAL results using the backbone LLM
>
> We acknowledge that CoT and PAL are both based on LLMs. That being said, CoT and PAL are quite different and should be treated as two different models with distinct behaviors as argued in the original paper. We believe that the simplicity and effectiveness of combining models with different prompts should be attributed to the versatile nature of LLMs. This is a feature, not a bug.
>
> >  It would be beneficial to conduct further experiments with genuinely distinct models to truly harness their respective strengths, such as integrating CodeX and ChatGPT with GPT-4.
>
> | Base 1              | Base 2             | Selector Backbone | Accuracy | Success Selection | Upper Bound |
> | ------------------- | ------------------ | ----------------- | -------- | ----------------- | ----------- |
> | [Codex]   PAL 71.5% | [ChatGPT] CoT 80.8 | [ChatGPT]           | 81.2     | 70.6              | 87.8        |
> | [Codex] PAL 71.5    | [ChatGPT] CoT 80.8 | [GPT-4]             | 84.4     | **84.8**          | 87.8        |
> | [ChatGPT] PAL 79.2  | [ChatGPT] CoT 80.8 | [ChatGPT]           | **82.6** | 60.4              | **89.4**    |
> | [GPT-4] PAL 94.0    | [GPT-4] CoT 94.6   | [GPT-4]             | **95.6** | 72.6              | **97.1**    |
>
> We experimented with different backbone LLMs. Using our approach, when integrating PAL outputs from CodeX and CoT outputs from ChatGPT, employing GPT-4 as a model selector, we got a notable 3.2% performance boost. This not only showcases the versatility of our method but also hints at the potential benefits when integrating models tailored for specialized tasks. Our approach would be effective for sets of questions necessitating multiple models fine-tuned for specialized domains/tasks.
>
> > The proposed method demonstrates effectiveness, but it necessitates thrice the computational and inferential resources for a single problem.
>
> (Below) Computation cost with **ChatGPT** on GSM8K. #Tokens = #Input Tokens + #Generated Tokens. Cost is calculated based on [OpenAI Pricing](https://openai.com/pricing).
>
> | Method      | Accuracy | Cost(USD) | #Tokens    | #Gen. Tokens |
> | ----------- | -------- | --------- | ---------- | ------------ |
> | CoT+SC@5    | 85.4     | 11.23     | 7.19M      | 906K         |
> | CoT+SC@10   | 86.7     | 22.46     | 14.37M     | 1.81M        |
> | CoT+SC@15   | 87.4     | 33.69     | 21.56M     | 2.72M        |
> | CoT+SC@20   | 87.4     | 44.92     | 28.74M     | 3.62M        |
> | PAL+SC@5    | 80.9     | 13.88     | 9.04M      | 632K         |
> | PAL+SC@10   | 82.6     | 27.76     | 18.09M     | 1.27M        |
> | PAL+SC@15   | 82.4     | 41.64     | 27.13M     | 1.90M        |
> | PAL+SC@20   | 82.7     | 55.52     | 36.18M     | 2.54M        |
> | **Ours @5** | **88.2** | **29.29** | **18.99M** | **1.61M**    |
>
> (Below) Computation cost with **GPT-4** on GSM8K. #Tokens = #Input Tokens + #Generated Tokens. Cost is calculated based on OpenAI Pricing.
>
> | Method      | Accuracy | Cost(USD) | #Tokens    | #Gen. Tokens |
> | ----------- | -------- | --------- | ---------- | ------------ |
> | CoT+SC@5    | 95.6     | 183       | 5.21M      | 891K         |
> | CoT+SC@10   | 95.5     | 367       | 10.43M     | 1.79M        |
> | CoT+SC@15   | 95.8     | 550       | 15.64M     | 2.68M        |
> | CoT+SC@20   | 95.8     | 733       | 20.85M     | 3.57M        |
> | PAL+SC@5    | 94.7     | 173       | 5.13M      | 633K         |
> | PAL+SC@10   | 95.4     | 346       | 10.27M     | 1.27M        |
> | PAL+SC@15   | 95.5     | 519       | 15.4M      | 1.90M        |
> | PAL+SC@20   | 95.8     | 692       | 20.53M     | 2.53M        |
> | **Ours @5** | **96.5** | **386**   | **11.33M** | **1.54M**        |
>
> It's pivotal to consider the entire computational landscape: we actually reduce computational overhead, when looking at multi-path inference (i.e., high accuracy regime). As illustrated in our results, our approach's computational cost is marginally higher than that of CoT+SC@10 and PAL+SC@10, but lower than CoT@15 and PAL@15. Notably, our approach even outperforms CoT@20 and PAL@20, underscoring its efficiency.
>
> > The "model selection" issue as addressed in this paper seems to be a variant of the self-consistency methods, a topic that has been extensively explored. Prioritizing the selection of prompt strategies based on a specific problem might present a more significant research avenue.
>
> Self-consistency was proposed to address the errors introduced in greedy decoding or random sampling, while our method is proposed to combine systems with different strengths and weaknesses. As shown in Table 2 of our paper, using our method with only 5 samples, outperforms using self-consistency with 15 samples which use roughly the same computation. In particular, with GPT-4, we achieve an accuracy of 96.5 using 5 samples while the baseline CoT with 15 samples achieves an accuracy of 95.8. Similarly, with ChatGPT, we achieve 88.2 while CoT achieves 87.4.
>
> Moreover, as shown in the table in the previous question, the performance of vanilla Self-Consistency tends to plateau after 15 paths. Our method, by selecting between two reasoning models, transcends this performance ceiling and achieves state-of-the-art results on GSM8K with 96.8% accuracy, underpinning its effectiveness and superiority over vanilla Self-Consistency.
>
> > The motivation behind the paper stems from the notion that CoT and PAL excel in different task types. However, a thorough analysis and explicit examples highlighting the disparities between these two methods are conspicuously absent. Furthermore, there's a need for elucidation on why these methods showcase different correct distributions within the experimental section.
>
> We acknowledge that a more in-depth analysis of CoT and PAL would be helpful. We manually analyze 100 data instances in GSM8K where the results of CoT and PAL disagree (50 CoT correct PAL incorrect, 50 CoT incorrect PAL correct). After filtering some ambiguous instances, we got the below table.
> | Mistake Categories                         | CoT Failure Cases | PAL Failure Cases |
> | ------------------------------------------ | ----------------- | ----------------- |
> | Computation Error                          | **5**             | 0                 |
> | Problem Decomposition & Logic Relationship | **24**            | 5                 |
> | Flexibility                                | 1                 | **13**            |
> | Semantic Understanding                     | 8                 | **20**            |
> | Context Awareness                          | 5                 | **9**             |
>
> 1. Computation Error:  PAL: Excels in computation accuracy, avoiding precision errors. CoT: Tends to falter in complex computations.
> 2. Problem Decomposition & Logic Relationship: PAL: Breaks down problems step-by-step, leveraging variables to map logical relations, thus simplifying tasks for LLMs. CoT: Might grapple with intricate logical relationships
> 3. Flexibility: CoT: Offers greater flexibility, enabling both forward and backward logical reasoning, sometimes solving through equation methodologies. PAL: Does not perform well on problems involving backward reasoning.
> 4. Semantic Understanding: CoT: Utilizes natural language prowess for superior semantic understanding. PAL: At times, misinterprets problem statements.
> 5. Context Awareness: CoT: Allows for a more holistic view of the problem, taking into account all parts of the narrative and not just individual operations. PAL: May decompose the problem too carefully and focus on the local operation.
>
> Furthermore, as shown in the Section 4.1 of the paper, by comparing the upper bound of CoT-PAL and CoT-CoT’, we show from the side that the correct distribution of CoT-PAL is more different than that of CoT-CoT’, underscoring the unique strengths each method brings to problem-solving. We will add a detailed analysis with showcases in our next version.
>
> > Regarding the PAL results, are the final outcomes integrated into the selection model, or is it limited to the generated program alone?
>
> Thanks for pointing it out. In our approach, the final outcomes generated by PAL are not integrated into the selection model. Instead, we only use the generated Python program as shown in Figure 1 of the paper. We also conducted a supplemental experiment on GSM8K with ChatGPT, appending the sentence "So the result is {PAL result}" to the program. After running this experiment three times, the results showed a mean accuracy of 82.6±0.08. It is almost the same as the result 82.6%, without integrating the final outcome. This indicates that incorporating the final outcome has a negligible impact on the model's overall performance.
>
> > Does this model exclusively employ its "self-debug" feature for evaluating these results?
>
> Whether the model is self-debugging depends on subjective judgment. The selection model likely takes several aspects into account, which might include bugs. However, we observed that Python programs generated for these reasoning tasks are relatively straightforward. Therefore, they are unlikely to contain bugs in the sense of software engineering that would require debugging.

---

### Meta-Review · Area_Chair_DuTz · 2023-09-25

**Recommendation:** 3

**Metareview:**

This paper explores how to combine two different reasoning methods, Chain-of-Thought (CoT) and Program-Aided-Language Models (PAL), for various inference tasks. The paper uses a large language model (LLM) to dynamically select between the outputs of CoT and PAL, based on two factors: the complexity of the problem and the confidence of the methods. The paper evaluates the proposed method on eight inference datasets, using different LLMs such as Codex, ChatGPT, and GPT-4. The paper achieves state-of-the-art results on two datasets, GSM8K and SVAMP, and provides theoretical and empirical analysis of the method’s effectiveness. Additionally the proposed method is complementary to the self-consistency (a prompting technique) and can be combined with it to further improve the performance. The authors have also been able to justify the concerns raised by the reviewers about the high inference cost of the proposed approach.

---

### Decision · Program_Chairs · 2023-10-07

**Decision:**

Accept-Findings

**Comment:**

This paper explores how to combine two different reasoning methods, Chain-of-Thought (CoT) and Program-Aided-Language Models (PAL), for various inference tasks. The paper uses a large language model (LLM) to dynamically select between the outputs of CoT and PAL, based on two factors: the complexity of the problem and the confidence of the methods. The paper evaluates the proposed method on eight inference datasets, using different LLMs such as Codex, ChatGPT, and GPT-4. The paper achieves state-of-the-art results on two datasets, GSM8K and SVAMP, and provides theoretical and empirical analysis of the method’s effectiveness. Additionally the proposed method is complementary to the self-consistency (a prompting technique) and can be combined with it to further improve the performance. The authors have also been able to justify the concerns raised by the reviewers about the high inference cost of the proposed approach.